# Not all Medicaid for pregnancy care is delivered equally

**Jonas J. Swartz**[1]\*, **Menolly Kaufman**[2,3], **Maria I. Rodriguez**[4]

**1** Department of Obstetrics and Gynecology, Division of Women's Community and Population Health, Duke University School of Medicine, Durham, NC, United States of America, **2** Center for Health Systems Effectiveness, Oregon Health & Science University, Portland, OR, United States of America, **3** Oregon Health & Science University-Portland State University School of Public Health, Portland, OR, United States of America, **4** Department of Obstetrics and Gynecology, Division of Complex Family Planning, Oregon Health & Science University, Portland, OR, United States of America

\* jonas.swartz@duke.edu

## Abstract

### Objectives

Pregnant beneficiaries in the two primary Medicaid eligibility categories, traditional Medicaid and pregnancy Medicaid, have differing access to care especially in the preconception and postpartum periods. Pregnancy Medicaid has higher income limits for eligibility than traditional Medicaid but only provides coverage during and for a limited time period after pregnancy. Our objective was to determine the association between type of Medicaid (traditional Medicaid and pregnancy Medicaid) on receipt of outpatient care during the perinatal period.

### Methods

This retrospective cohort study compared outpatient visits using linked birth certificate and Medicaid claims from all Medicaid births in Oregon and South Carolina from 2014 through 2019. Pregnancy Medicaid ended 60 days postpartum during the study. Our primary outcome was average number of outpatient visits per 100 beneficiaries each month during three perinatal time points: preconceputally (three months prior to conception), prenatally (9 months prior to birthdate) and postpartum (from birth to 12 months).

### Results

Among 105,808 Medicaid-covered births in Oregon and 141,385 births in South Carolina, pregnancy Medicaid was the most prevalent categorical eligibility. Traditional Medicaid recipients had a higher average number of preconception, prenatal and postpartum visits as compared to those in pregnancy Medicaid.

### Discussion

In South Carolina, those using traditional Medicaid had 450% more preconception visits and 70% more postpartum visits compared with pregnancy Medicaid. In Oregon, those using traditional Medicaid had 200% more preconception visits and 29% more postpartum visits than individuals using pregnancy Medicaid. Lack of coverage in both the preconception and

**Data Availability Statement:** Data are available from the Oregon Health Authority and South Carolina Department of Health and Human Services under data use agreement provisions. Per the data use agreement, the relevant limited data sets cannot be made publicly available. Medicaid

data from Oregon can be requested from the Oregon Health Authority by scheduling a consultation via email at OHA. HealthAnalyticsRequest@odhsoha.oregon.gov. South Carolina's Department of Health and Human Services directs research inquiries to call South Carolina Revenue and Fiscal Affairs Office, Data Integration and Analysis Division, (803) 734-3793.

**Funding:** This work was conducted with the support of award 1R01MD013648-01 for MIR from the National Institute on Minority Health and Health Disparities (https://www.nimhd.nih.gov/) and grant K12HD103083 which supports JJS from the National Institute of Child Health and Human Development (NICHD) (https://www.nichd.nih.gov/) of the U.S. National Institutes of Health. The content is solely the responsibility of the authors and does not necessarily represent the official views of the National Institutes of Health. The funders had no role in study design, data collection and analysis, decision to publish, or preparation of the manuscript. There was no additional external funding received for this study.

**Competing interests:** The authors have declared that no competing interests exist.

postpartum period deprive women of adequate opportunities to access health care or contraception. Changes to pregnancy Medicaid, including extended postpartum coverage through the American Rescue Plan Act of 2021, may facilitate better continuity of care.

## Introduction

Medicaid pays for over 40% of obstetric care in the United States (US) [1]. Access to Medicaid insurance is associated with increased likelihood of preconception coverage, decreased probability of periods of uninsurance during pregnancy, and increased likelihood of postpartum coverage [2,3]. Pregnant people with Medicaid also frequently have changes in their insurance type before and after pregnancy [4]. Prior studies of perinatal Medicaid populations often grouped those with all categories of Medicaid compared to no insurance or private insurance [2,3]. States have multiple categories of Medicaid programs enrolling pregnant people. Recognizing the importance of obstetrical care, all states but South Dakota cover pregnant people through pregnancy Medicaid at higher income thresholds than non-pregnant adults [5]. Pregnancy Medicaid generally includes access to comprehensive coverage and lasts from pregnancy diagnosis through 60 days postpartum while traditional Medicaid is comprehensive coverage irrespective of pregnancy.

Our objective was to determine the association between type of Medicaid (traditional Medicaid and pregnancy Medicaid) on Medicaid enrollment and receipt of outpatient care during the 12 months before and after a birth.

## Methods

This retrospective cohort study compared outpatient visits using linked birth certificate and Medicaid claims from all Medicaid live births in two states (Oregon and South Carolina). Our study period included January 2014- December 2019.

Pregnant women in each state meeting citizenship and financial criteria used traditional Medicaid or pregnancy Medicaid. Emergency Medicaid recipients have more limited coverage and were excluded [6]. Each state has multiple programs unrelated to pregnancy that offer categorical eligibility for services (e.g. adults with dependent children, blindness or disability), provided enrollees are below income thresholds. These multiple eligibility categories comprise the traditional Medicaid population in this study. In Oregon, adults earning up to 138% of the Federal Poverty Level (FPL) may enroll in traditional Medicaid, while in South Carolina, they may earn up to 67% of FPL. Both states offer expanded coverage for individuals during pregnancy (190% of FPL in Oregon and 199% of FPL in South Carolina). During our study period, pregnancy Medicaid ended at 60 days postpartum. Despite this coverage limit, some beneficiaries may maintain coverage for longer periods postpartum or gain other categorical coverage that appears in claims data.

Our primary outcome was average number of outpatient visits per 100 beneficiaries each month during three perinatal time points: preconceputally (three months prior to conception), prenatally (9 months prior to birthdate) and postpartum (from birth to 12 months). We used Medicaid claims rather than birth certificate data to identify prenatal visits because birth certificates report only the total number of visits, not the timing of visits during the pregnancy. We measured utilization of outpatient care during pregnancy as the average number of visits per 100 beneficiaries each month. We also measured the number of months of enrollment at each perinatal time point. We conducted sensitivity analyses evaluating binary indicators for at least

one preconception visit, at least seven prenatal visits, at least one postpartum visit in the first 60 days postpartum, and at least one visit from 61–365 days postpartum.

We assessed for differences in utilization in the preconception, prenatal and postpartum periods using analysis of covariance (ANCOVA) allowing us to adjust for maternal age, parity, rurality, history of cesarean delivery, hypertensive disease of pregnancy and diabetes during pregnancy. All analyses were completed in R 4.10. The Institutional Review Board at Oregon Health & Science University approved this study. Administrative data sources were used not requiring consent. Data cannot be made available per data use agreements with Oregon and South Carolina.

## Results

We observed 105,808 Medicaid-covered births in Oregon and 141,385 births in South Carolina from 2014 to 2019. Pregnancy Medicaid covered the majority of pregnancies in each state (Oregon 58,044 [54.9%], South Carolina 86,472 [61.1%]) (Table 1). The states had similar distribution of rural versus urban residents among the birthing population as well as similar occurrence of common comorbidity such as hypertension and diabetes.

In both Oregon and South Carolina, traditional Medicaid recipients had a higher average number of preconception, prenatal and postpartum visits compared to those in pregnancy Medicaid. (Table 2 and Fig 1). Preconception visits were infrequent in Oregon among recipients of pregnancy and traditional Medicaid, after adjusting for age, rurality, multiparous, history of previous cesarean, gestational hypertension, and gestational diabetes. Notably, however, those with traditional Medicaid had 200% more preconception visits than those with pregnancy Medicaid (mean 0.2 visits (SE 0.00) vs 0.1 visits (SE 0.00)). In South Carolina, those with traditional Medicaid had 450% more preconception visits than those with pregnancy Medicaid (mean 0.9 visits (SE 0.01) vs mean 0.2 visits (SE 0.01)). Results were similar in a sensitivity analysis conducted with dichotomized indicators of preconception, prenatal and

**Table 1. Demographic characteristics by medicaid type and perinatal period, Oregon and South Carolina 2014–2019.**

| | Enrollees | |
| --- | --- | --- |
| | **Oregon**<br>**N (%)** | **South Carolina**<br>**N (%)** |
| **Medicaid Type** | | |
| Emergency Medicaid | 16,115 (13.2) | 12,641 (8.4) |
| Pregnancy Medicaid | 58,044 (54.9) | 86,472 (61.1) |
| Traditional Medicaid | 47,764 (45.1) | 54,913 (38.8) |
| **Maternal Age (mean (SD))** | 27.2 (5.8) | 25.8 (5.6) |
| **Maternal Residential Location** | | |
| Urban | 92,021 (75.5) | 105,387 (69.8) |
| Rural | 27,050 (22.2) | 44,032 (29.2) |
| Missing | 2,852 (2.3) | 1,608 (1.1) |
| **Multiparous** | 79,806 (65.5) | 93,528 (61.9) |
| **Prior cesarean** | 17,622 (14.5) | 24,792 (16.4) |
| **Prepregnancy hypertension** | 2,526 (2.1) | 4,934 (3.3) |
| **Gestational hypertension** | 8,754 (7.2) | 11,761 (7.8) |
| **Prepregnancy diabetes** | 14,31 (1.2) | 1,675 (1.1) |
| **Gestational diabetes** | 10,635 (8.7) | 9,076 (6.0) |

Abbreviations: SD = Standard Deviation.

**Table 2. Population and perinatal outpatient utilization by medicaid type and perinatal period, Oregon and South Carolina 2014–2019.**

| | Unadjusted | | Adjusted* | |
|---|---|---|---|---|
| | Oregon | South Carolina | Oregon | South Carolina |
| | Mean (SD) | Mean (SD) | Mean (SE) | Mean (SE) |
| **Preconception visits** | | | | |
| Pregnancy Medicaid | 0.1 (0.7) | 0.2 (1.0) | 0.1 (0.00) | 0.2 (0.01) |
| Traditional Medicaid | 0.2 (1.0) | 0.9 (2.8) | 0.2 (0.00) | 0.9 (0.01) |
| **Prenatal Visits** | | | | |
| Pregnancy Medicaid | 14.2 (18.3) | 16.6 (11.7) | 14.5 (0.08) | 16.0 (0.04) |
| Traditional Medicaid | 17.0 (22.8) | 18.5 (14.5) | 17.1 (0.09) | 18.7 (0.05) |
| **Visits (up to one year after delivery)** | | | | |
| Pregnancy Medicaid | 8.2 (24.4) | 4.1 (9.3) | 8.3 (0.10) | 4.0 (0.04) |
| Traditional Medicaid | 10.8 (29.3) | 6.6 (12.4) | 10.7 (0.11) | 6.7 (0.05) |
| **Visits in the first 60 days after delivery** | | | | |
| Pregnancy Medicaid | 2.0 (5.3) | 1.7 (3.3) | 2.0 (0.02) | 1.7 (0.01) |
| Traditional Medicaid | 2.4 (6.3) | 1.9 (3.4) | 2.4 (0.02) | 1.9 (0.1) |
| **Visits 61–365 days after delivery** | | | | |
| Pregnancy Medicaid | 6.2 (20.1) | 2.3 (7.5) | 6.3 (0.08) | 2.3 (0.03) |
| Traditional Medicaid | 8.3 (24.1) | 4.8 (10.4) | 8.3 (0.09) | 4.8 (0.04) |

*Adjusted for age, rurality, multiparous, history of previous cesarean, gestational hypertension, and gestational diabetes.

SD = Standard Deviation.

SE = Standard Error.

postpartum care demonstrating a higher percentage of patients with traditional Medicaid had at least one-preconception visit than those with pregnancy Medicaid in both states (S1 Table). A higher proportion of traditional Medicaid recipients also had at least seven prenatal visits than those with pregnancy Medicaid in both states.

In Oregon, pregnancy and traditional Medicaid recipients had similar mean months of enrollment in the preconception (pregnancy 1.2 months vs traditional 1.5 months), prenatal (pregnancy 8.9 months vs traditional 9.1 months) and 61 day to 365 day postpartum periods (pregnancy 7.4 months vs traditional 8.0 months). In South Carolina, pregnancy Medicaid recipients had fewer mean months of enrollment in the preconception (pregnancy 0.5 months vs 1.5 months), prenatal (pregnancy 7.3 months vs traditional 10.4 months) and 61 day to 365 day postpartum periods (pregnancy 5.2 months vs traditional 8.0 months) (S1 Table).

Care access postpartum was also different depending on Medicaid eligibility, especially from 61–365 days postpartum. In Oregon those with traditional Medicaid had 29% more 61–365 days postpartum as those with pregnancy Medicaid (mean 8.3 visits (SE 0.09) vs 6.3 visits (SE 0.08)). In South Carolina, those with traditional Medicaid had 70% more visits 61–365 days postpartum as those with pregnancy Medicaid (mean 4.8 visits (SE 0.04) vs mean 2.3 visits (SE 0.03)).

## Discussion

Lack of coverage creates stark barriers to access among those who lose insurance. Pregnancy Medicaid was associated with a marked decrease in preconception visits and postpartum visits compared with traditional Medicaid in South Carolina where access to Medicaid outside of pregnancy is limited. In Oregon, those with pregnancy Medicaid also were less likely to access preconception or postpartum care than those with traditional Medicaid. These utilization

### a.    Oregon

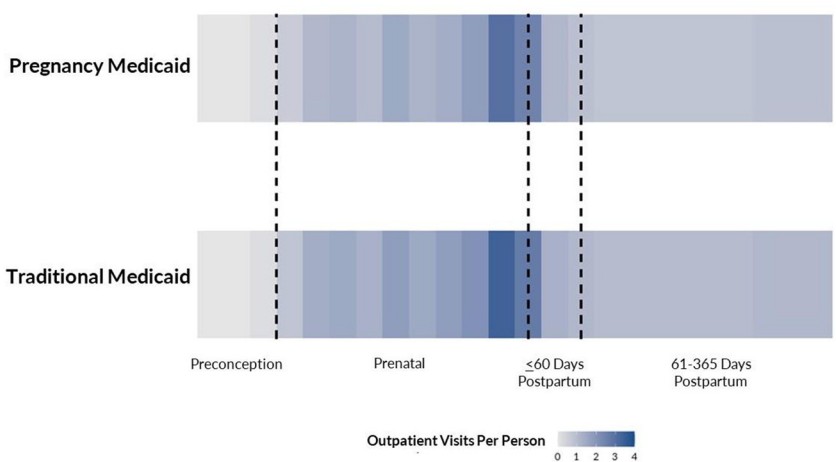

### b.   South Carolina

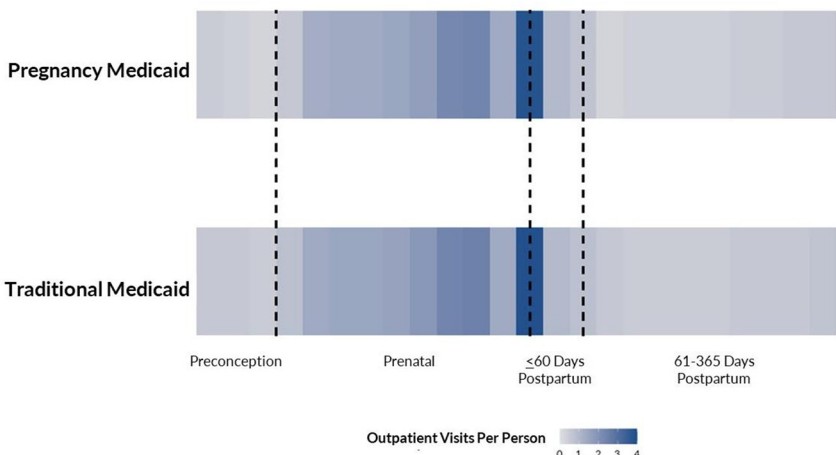

**Fig 1. Perinatal outpatient utilization by medicaid type, Oregon & South Carolina 2014–2019.** Outpatient visits among two Medicaid eligibility categories in Oregon (panel a) and South Carolina (panel b). Dashed lines demarcate the preconception period, delivery and 60 days after delivery as, during the study period, pregnancy Medicaid coverage ended at this time postpartum. Each vertical rectangle in the panel represents average number of outpatient visits per person per month. Darker color indicates and higher number of average outpatient visits during that month.

patterns were shaped by differences in temporal patterns of enrollment with traditional Medicaid recipients more likely to be enrolled in Medicaid in time periods before and after pregnancy.

Preconception visits allow individuals with chronic conditions to avoid teratogenic exposures and optimize conditions like diabetes that may affect maternal and fetal health during pregnancy [7]. Postpartum care provides an essential opportunity to treat chronic conditions identified during pregnancy, establish primary care, and address contraception [8]. If pregnancy is a window to future health, lack of coverage in both the preconception and postpartum

period deprives individuals of adequate opportunities improve outcomes during that window or in the future [8].

Limitations of this analysis include use of administrative data, inability to identify care utilization through other payors, and lack of generalizability of data from two states. We also did not analyze timing of initiation of prenatal care, though that would be an opportunity for future research. However, data from Wisconsin demonstrated a similar lower rate of postpartum care attendance among those eligible for pregnancy-only Medicaid, suggesting this finding likely would extend to other states [9]. Some difference in care utilization may be attributable to differences in the individuals eligible for traditional Medicaid versus pregnancy Medicaid. Those with traditional Medicaid have lower incomes and could have more chronic conditions that could require increased visits, for example. Claims data also does not capture care obtained but not financed by Medicaid, so individuals could have received care unobserved in this dataset outside the time of their Medicaid enrollment. Additionally, because maternity care is often bundled in global billing for delivery, we may have under-identified prenatal and postpartum visits.

Given the high proportion of births in the United States covered by Medicaid, policies expanding or restricting access to Medicaid have tremendous potential to affect maternal health outcomes. Income eligibility for pregnancy Medicaid is generally higher than for traditional Medicaid, for example up to 190% of the federal poverty level (FPL) in Oregon, 199% of FPL in South Carolina, and a high of 380% of FPL in Iowa.[5] In contrast, income eligibility for traditional Medicaid is up to 138% of FPL in Oregon and a paltry 67% of FPL in South Carolina [10]. In both states, this income eligibility discrepancy puts beneficiaries at risk of being ineligible for Medicaid coverage when they are not pregnant. Traditional Medicaid may be a favorable program for pregnant Medicaid beneficiaries because they are not at risk of losing eligibility after pregnancy. Data from North Carolina indicate a larger proportion of those with pregnancy Medicaid were at risk of losing coverage by four months postpartum [11]. Additionally, some states limit the scope of covered services under pregnancy Medicaid as compared to traditional Medicaid plans [12].

The American Rescue Plan of 2021 allows states to extend pregnancy Medicaid coverage to include 12 months of postpartum coverage [13]. As of April 2023, 46 states plan to offer care through this expansion which may increase parity with the traditional Medicaid care utilization observed in this study [14]. This policy at least delays the financial cliff between pregnancy and traditional Medicaid eligibility. Time limits in enrollment constrict utilization of care and, conversely, policies like expanded postpartum care or expansion of Medicaid under the Affordable Care Act can have multiple positive effects. Research on the effects of the Affordable Care Act on insurance churn and perinatal insurance coverage demonstrate decreased uninsurance, increased utilization of care, and, in some studies, reduction in adverse birth outcomes concentrated among those with highest risk [15–18].

## Conclusion

Medicaid beneficiaries historically had disparate access to care, particularly in the preconception and postpartum periods, depending on their Medicaid eligibility category. Medicaid eligibility drives access to care. Pregnancy Medicaid offers program eligibility at higher income levels, but with reduced access to preconception and postpartum care. Extended postpartum coverage for this eligibility category through the American Rescue Plan 2021 may help improve perinatal health. Research on these newly implemented programs should include assessment of access to preconception care, as that gap may not be addressed by expanded postpartum access.

## Supporting information

**S1 Table. Medicaid enrollment and utilization by medicaid type and perinatal period, Oregon and South Carolina 2014–2019.** This table presents data on Medicaid enrollment and visits by beneficiaries in the perinatal period in Oregon and South Carolina.
(DOCX)

**S1 File.**
(DOCX)

## Author Contributions

**Conceptualization:** Jonas J. Swartz.

**Data curation:** Menolly Kaufman.

**Formal analysis:** Menolly Kaufman.

**Funding acquisition:** Maria I. Rodriguez.

**Investigation:** Maria I. Rodriguez.

**Methodology:** Jonas J. Swartz, Menolly Kaufman.

**Resources:** Maria I. Rodriguez.

**Software:** Menolly Kaufman.

**Supervision:** Maria I. Rodriguez.

**Visualization:** Jonas J. Swartz, Menolly Kaufman, Maria I. Rodriguez.

**Writing – original draft:** Jonas J. Swartz.

**Writing – review & editing:** Jonas J. Swartz, Menolly Kaufman, Maria I. Rodriguez.

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
