## [Decision Letter · Decision Letter 0]

6 Sep 2023

PONE-D-23-04247Not all Medicaid for pregnancy care is delivered equallyPLOS ONE

Dear Dr. Swartz,

Thank you for submitting your manuscript to PLOS ONE. After careful consideration, we feel that it has merit but does not fully meet PLOS ONE’s publication criteria as it currently stands. Therefore, we invite you to submit a revised version of the manuscript that addresses the points raised during the review process.

The manuscript has been evaluated by two reviewers, and their comments are available below.

The reviewers have raised a number of major concerns. They feel the manuscript should outline a clearly-defined research question, and they request improvements to the reporting of methodological aspects of the study, for example, regarding the exclusion criteria and more information on how the data collection was completed. The reviewers also note concerns about the statistical analyses presented and request re-analyses be completed.

Could you please carefully revise the manuscript to address all comments raised?

We look forward to receiving your revised manuscript.

Kind regards,

Avanti Dey, PhD

Senior Staff Editor

PLOS ONE

Journal Requirements:

“This work was conducted with the support of award 1R01MD013648-01 for MIR from the National Institute on Minority Health and Health Disparities (https://www.nimhd.nih.gov/) and grant K12HD103083 for JJS from the National Institute of Child Health and Human Development (NICHD) (https://www.nichd.nih.gov/) of the U.S. National Institutes of Health. The content is solely the responsibility of the authors and does not necessarily represent the official views of the National Institutes of Health.

Reviewers' comments:

Reviewer's Responses to Questions

**Comments to the Author**

1. Is the manuscript technically sound, and do the data support the conclusions?

Reviewer #1: Partly

Reviewer #2: Yes

2. Has the statistical analysis been performed appropriately and rigorously? 

Reviewer #1: I Don't Know

Reviewer #2: Yes

3. Have the authors made all data underlying the findings in their manuscript fully available?

Reviewer #1: No

Reviewer #2: Yes

4. Is the manuscript presented in an intelligible fashion and written in standard English?

Reviewer #1: Yes

Reviewer #2: Yes

5. Review Comments to the Author

Reviewer #1: This paper examines rates of outpatient care before, during, and after pregnancy among those enrolled in Medicaid via the pregnancy pathway versus all other pathways. They find that those who were enrolled in Medicaid via the pregnancy pathway had lower rates of preconception, prenatal, and postpartum visits than those in traditional Medicaid using linked claims and birth record data from Oregon and South Carolina. The study uses unique data and addresses an important and understudied topic, but could benefit from additional clarification and detail.

Major:

1. This study frames the outcomes as “access to care” in the perinatal periods, but this is a secondary outcome to what is driving access to care in this context, which is eligibility for Medicaid. Those who are eligible for Medicaid only due to pregnancy are by definition not eligible before or after pregnancy based on their incomes, so a large portion of those in this group will fall out of observation. It is unclear how the study’s methodology deals with this. Does the denominator of the rates include those who fall out of coverage or exclude them? It is therefore unclear whether the study is comparing access to care within Medicaid enrollees or comparing those who maintain coverage to those who drop out and care can therefore not be observed.

a. If the rates includes those who drop out of coverage, they are comparing utilization of services among those who they have data for to those who are lost to follow-up due to lack of eligibility for the program. This makes the results difficult to interpret since a lack of outpatient care prior to or after pregnancy does not mean that no outpatient care was received, it just wasn’t paid for by Medicaid and therefore is unobservable in the data. This is an important distinction to make clearly and up front. This renders the findings less relevant because we would expect that those who are not in Medicaid would not have any Medicaid-financed care.

b. However, if the denominator restricts to those who are enrolled due to pregnancy but also maintain their coverage prenatally and postpartum, this would imply that there is something about being enrolled in the pregnancy pathway specifically that results in lower outpatient use (and the authors should spend more time unpacking and hypothesizing what this is). However, if the authors do condition on enrollment, then the timing of assessment of eligibility pathway becomes highly relevant. If someone is enrolled in the pregnancy pathway during pregnancy but enrolled as a low-income parent after pregnancy, which group are they classified in? Are people switching eligibility pathway to pregnancy when they become pregnant? At what time point during pregnancy was eligibility pathway assessed?

2. Relatedly, those who remain in Medicaid before and after pregnancy may be systematically different from those who are only enrolled due to pregnancy. This is because income eligibility limits for pregnancy are higher than for other groups, and therefore those who maintain insurance beyond pregnancy are typically lower income which may be associated with health status and other factors.

3. Were outpatient prenatal visits measured using the birth record or the claims? Prenatal care in the birth record would capture prenatal care not covered via Medicaid which would be one way to get around the issue of lack of observation of non-Medicaid financed care. Otherwise you don’t actually know that those who are not in the Medicaid program preconception or postpartum aren’t getting outpatient care via other insurance.

4. Did the study restrict to the pregnant population or to those who had claims that indicated a delivery/labor?

5. The study groups eligibility into pregnancy versus traditional, but there are many different reasons for Medicaid eligibility beyond pregnancy which have different implications for utilization of services (i.e., disability, low-income adults, parents, children, etc). How were these grouped? What reasons for eligibility are categorized as “traditional”?

6. More specific language needed in first paragraph – what is the difference between full benefits and comprehensive coverage?

7. How are you defining the perinatal period? There are specific definitions of this term so would suggest avoiding use of this term and just defining how you measured this period.

8. Results section is brief and does not describe differences by subgroups or across the different time points. This section reports the average number of visits but the outcome was assessed as a rate.

Minor:

1. The specific numerical results in the discussion section would fit better in the results section.

2. In the abstract it says that pregnancy-related Medicaid eligibility ends 60 days postpartum. This is no longer the case in the majority of states due to postpartum Medicaid extensions.

3. Medicaid is the most common public payer but commercial is largest payer of births in the U.S.

Reviewer #2: I would like to thank the authors for giving me the opportunity to review this paper.

The paper titled "Not all Medicaid for pregnancy care is delivered equally" by Swartz et al investigates the association between Medicaid types (Pregnancy Medicaid and Traditional Medicaid) and outpatient care utilization during the perinatal period in Oregon and South Carolina from 2014 to 2019.

The paper addresses an essential topic concerning disparities in healthcare access for pregnant Medicaid beneficiaries and presents valuable insights. Strengths of this study are:

• Addresses an important and relevant topic related to healthcare disparities among pregnant Medicaid beneficiaries.

• Utilizes a retrospective cohort design, linked birth certificate, and Medicaid claims data, enhancing the study's validity and robustness.

• Presents clear and easy-to-understand tables, including unadjusted and adjusted means, allowing for comprehensive interpretation of outpatient care utilization during the perinatal period.

• Effectively emphasizes the significance of preconception and postpartum care in improving maternal and fetal health outcomes.

• Acknowledges the potential implications of lacking coverage during critical periods, such as missed opportunities for accessing healthcare and contraception.

• Provides valuable insights into outpatient care utilization for both Medicaid types (Pregnancy Medicaid and Traditional Medicaid) in two states (Oregon and South Carolina) from 2014 to 2019.

• Highlights the importance of considering additional adjustment variables to account for potential confounding factors that could influence the study's findings.

• Recommends the inclusion of data from more states with diverse demographics and healthcare systems to enhance the generalizability of the results.

• Acknowledges the policy changes related to extended postpartum Medicaid coverage and suggests the need for further research in this area.

The paper's findings can contribute to policy discussions and potential positive changes in maternal healthcare access and outcomes for Medicaid beneficiaries.

In summary, this is a well-written and concise paper and I would recommend it for publication with some minor changes as follows:

1. Please provide a concise conclusion section to this paper which can help the reader to gain the main points of the study in a single glance.

2. Line 104 mentions “ANCOVA…”. Do you mean analysis of variances (ANOVA)? Please clarify this to the reader.

6. PLOS authors have the option to publish the peer review history of their article (what does this mean?). If published, this will include your full peer review and any attached files.

Reviewer #1: No

Reviewer #2: **Yes: **Muhammad Saqib

---

## [Author Response · Author response to Decision Letter 0]

10 Oct 2023

Dear PLOS editors

Thank you for the opportunity to revise this manuscript. We have included our response to reviewers and the editors below. 

Journal Requirements:

We made changes to fit the journal style.

“This work was conducted with the support of award 1R01MD013648-01 for MIR from the National Institute on Minority Health and Health Disparities (https://www.nimhd.nih.gov/) and grant K12HD103083 for JJS from the National Institute of Child Health and Human Development (NICHD) (https://www.nichd.nih.gov/) of the U.S. National Institutes of Health. The content is solely the responsibility of the authors and does not necessarily represent the official views of the National Institutes of Health.

We have updated the funding statement on the title page and in the cover letter. 

We moved the ethics statement. 

Reviewers' comments:

Reviewer's Responses to Questions

Comments to the Author

1. Is the manuscript technically sound, and do the data support the conclusions?

Reviewer #1: Partly

Reviewer #2: Yes

2. Has the statistical analysis been performed appropriately and rigorously? 

Reviewer #1: I Don't Know

Reviewer #2: Yes

3. Have the authors made all data underlying the findings in their manuscript fully available?

Reviewer #1: No

Reviewer #2: Yes

4. Is the manuscript presented in an intelligible fashion and written in standard English?

Reviewer #1: Yes

Reviewer #2: Yes

5. Review Comments to the Author

Reviewer #1: This paper examines rates of outpatient care before, during, and after pregnancy among those enrolled in Medicaid via the pregnancy pathway versus all other pathways. They find that those who were enrolled in Medicaid via the pregnancy pathway had lower rates of preconception, prenatal, and postpartum visits than those in traditional Medicaid using linked claims and birth record data from Oregon and South Carolina. The study uses unique data and addresses an important and understudied topic, but could benefit from additional clarification and detail.

Major:

1. This study frames the outcomes as “access to care” in the perinatal periods, but this is a secondary outcome to what is driving access to care in this context, which is eligibility for Medicaid. Those who are eligible for Medicaid only due to pregnancy are by definition not eligible before or after pregnancy based on their incomes, so a large portion of those in this group will fall out of observation. It is unclear how the study’s methodology deals with this. Does the denominator of the rates include those who fall out of coverage or exclude them? It is therefore unclear whether the study is comparing access to care within Medicaid enrollees or comparing those who maintain coverage to those who drop out and care can therefore not be observed.

We agree with the reviewer that one limitation of this methodology and dataset is that individuals could have unobserved care utilization that is not funded by Medicaid and, therefore, does not appear in claims. The denominator includes individuals enrolled in Medicaid at the time of birth. One opportunity for future research might be comparing enrollment patterns stratified by categorical eligibility. We added the following to the limitations to reflect the reviewer’s feedback: Line 202-204.

Claims data also does not capture care obtained but not financed by Medicaid, so individuals could have received care unobserved in this dataset outside the time of their Medicaid enrollment.

a. If the rates includes those who drop out of coverage, they are comparing utilization of services among those who they have data for to those who are lost to follow-up due to lack of eligibility for the program. This makes the results difficult to interpret since a lack of outpatient care prior to or after pregnancy does not mean that no outpatient care was received, it just wasn’t paid for by Medicaid and therefore is unobservable in the data. This is an important distinction to make clearly and up front. This renders the findings less relevant because we would expect that those who are not in Medicaid would not have any Medicaid-financed care.

We did include those who drop out of coverage, and have added the reviewer’s requested clarification to the limitations. (Line 202-204) We would suggest that the findings remain relevant because our objective is to characterize how different Medicaid programs shape access to care. While people may access care, prior data suggest that insurance lapses are common among women using Medicaid for pregnancy and that the most common pattern is periods of uninsurance rather than switching to private insurance.(1) Individuals may get care that is not captured during periods of uninsurance but it remains an impediment to care. 

b. However, if the denominator restricts to those who are enrolled due to pregnancy but also maintain their coverage prenatally and postpartum, this would imply that there is something about being enrolled in the pregnancy pathway specifically that results in lower outpatient use (and the authors should spend more time unpacking and hypothesizing what this is). However, if the authors do condition on enrollment, then the timing of assessment of eligibility pathway becomes highly relevant. If someone is enrolled in the pregnancy pathway during pregnancy but enrolled as a low-income parent after pregnancy, which group are they classified in? Are people switching eligibility pathway to pregnancy when they become pregnant? At what time point during pregnancy was eligibility pathway assessed?

We classify enrollment category based on enrollment at the time of delivery. Future research might be fruitful examining potential switching of eligibility pathways, as the reviewer suggests. However, that analysis is not included in this study. We suspect the primary driver of differential utilization is that pregnancy Medicaid recipients enroll later and disenroll earlier.

2. Relatedly, those who remain in Medicaid before and after pregnancy may be systematically different from those who are only enrolled due to pregnancy. This is because income eligibility limits for pregnancy are higher than for other groups, and therefore those who maintain insurance beyond pregnancy are typically lower income which may be associated with health status and other factors.

We agree with the reviewer that the populations are different between traditional and pregnancy Medicaid populations and that some of these differences may affect need for and use of care. We have added the following to the discussion: Line 199-202

Some difference in care utilization may be attributable to differences in the inidividuals eligible for Traditional Medicaid versus Pregnancy Medicaid. Those with Traditional Medicaid have lower incomes and could have more chronic conditions that could require increased visits, for example.

3. Were outpatient prenatal visits measured using the birth record or the claims? Prenatal care in the birth record would capture prenatal care not covered via Medicaid which would be one way to get around the issue of lack of observation of non-Medicaid financed care. Otherwise you don’t actually know that those who are not in the Medicaid program preconception or postpartum aren’t getting outpatient care via other insurance.

We used claims data for utilization outcomes. We have added acknowledgement of this to the limitations. (Line 202-204)

Claims data also does not capture care obtained but not financed by Medicaid, so individuals could have received care unobserved in this dataset outside the time of their Medicaid enrollment.

4. Did the study restrict to the pregnant population or to those who had claims that indicated a delivery/labor?

Analysis was restricted to those with live births. This is noted in the methods section: (Line 85)

This retrospective cohort study compared outpatient visits using linked birth certificate and Medicaid claims from all Medicaid live births in two states (Oregon and South Carolina).

5. The study groups eligibility into pregnancy versus traditional, but there are many different reasons for Medicaid eligibility beyond pregnancy which have different implications for utilization of services (i.e., disability, low-income adults, parents, children, etc). How were these grouped? What reasons for eligibility are categorized as “traditional”?

Traditional eligibility included, as suggested by the reviewer, multiple categories including those providing coverage for low-income, blind and disabled individuals. We grouped all of these programs, which were not specifically for pregnancy, as “traditional.” We intended to make the distinction that enrollment for these programs was not preconditioned on pregnancy. We include the following for clarification. (Line 94-97)

Each state has multiple programs unrelated to pregnancy that offer categorical eligibility for services (e.g. adults with dependent children, blindness or disability), provided enrollees are below income thresholds. These multiple enrollment categories comprise the traditional Medicaid population in this study.

6. More specific language needed in first paragraph – what is the difference between full benefits and comprehensive coverage?

Thank you for this clarification. We have updated the language to clarify that we intended to highlight the difference in duration of eligibility, not the services available in the two programs. (Line 76-78)

Pregnancy Medicaid generally includes access to comprehensive coverage and lasts from pregnancy diagnosis through 60 days postpartum while traditional Medicaid is comprehensive coverage irrespective of pregnancy.

7. How are you defining the perinatal period? There are specific definitions of this term so would suggest avoiding use of this term and just defining how you measured this period.

We are interested in the 12 months before and after a birth. Thank you for this request. We have updated the language. (Line 79-81)

Our objective was to determine the association between type of Medicaid (traditional Medicaid and pregnancy Medicaid) on receipt of outpatient care during the 12 months before and after a birth.

8. Results section is brief and does not describe differences by subgroups or across the different time points. This section reports the average number of visits but the outcome was assessed as a rate.

We expanded the results section to discuss the demographic characteristics of the study population (new Table 1). We also expanded our results section, as requested, and now includes both rates and absolute number of visits. 

Minor:

1. The specific numerical results in the discussion section would fit better in the results section.

These are now reported in the results section though some are also noted in the discussion when they are relevant. 

2. In the abstract it says that pregnancy-related Medicaid eligibility ends 60 days postpartum. This is no longer the case in the majority of states due to postpartum Medicaid extensions.

Thanks for this note. We discuss this at the end of the paper with updated data on adoption across the US. However, it was the case at the time of data collection. We changed the language to clarify. (Line 50)

Pregnancy Medicaid ended 60 days postpartum during the study.

3. Medicaid is the most common public payer but commercial is largest payer of births in the U.S.

We have made our language more precise.(Line 67)

Medicaid pays for over 40% of obstetric care in the United States (US)(2)

Reviewer #2

I would like to thank the authors for giving me the opportunity to review this paper.

The paper titled "Not all Medicaid for pregnancy care is delivered equally" by Swartz et al investigates the association between Medicaid types (Pregnancy Medicaid and Traditional Medicaid) and outpatient care utilization during the perinatal period in Oregon and South Carolina from 2014 to 2019. 

The paper addresses an essential topic concerning disparities in healthcare access for pregnant Medicaid beneficiaries and presents valuable insights. Strengths of this study are:

• Addresses an important and relevant topic related to healthcare disparities among pregnant Medicaid beneficiaries.

• Utilizes a retrospective cohort design, linked birth certificate, and Medicaid claims data, enhancing the study's validity and robustness.

• Presents clear and easy-to-understand tables, including unadjusted and adjusted means, allowing for comprehensive interpretation of outpatient care utilization during the perinatal period.

• Effectively emphasizes the significance of preconception and postpartum care in improving maternal and fetal health outcomes.

• Acknowledges the potential implications of lacking coverage during critical periods, such as missed opportunities for accessing healthcare and contraception.

• Provides valuable insights into outpatient care utilization for both Medicaid types (Pregnancy Medicaid and Traditional Medicaid) in two states (Oregon and South Carolina) from 2014 to 2019.

• Highlights the importance of considering additional adjustment variables to account for potential confounding factors that could influence the study's findings.

• Recommends the inclusion of data from more states with diverse demographics and healthcare systems to enhance the generalizability of the results.

• Acknowledges the policy changes related to extended postpartum Medicaid coverage and suggests the need for further research in this area.

The paper's findings can contribute to policy discussions and potential positive changes in maternal healthcare access and outcomes for Medicaid beneficiaries.

In summary, this is a well-written and concise paper and I would recommend it for publication with some minor changes as follows:

1. Please provide a concise conclusion section to this paper which can help the reader to gain the main points of the study in a single glance.

We added a conclusion. 

2. Line 104 mentions “ANCOVA…”. Do you mean analysis of variances (ANOVA)? Please clarify this to the reader.

Thank you for this question. We used analysis of covariance so that we could compare differences in means while controlling for covariates/confounders. We have updated the language in the document.(Line 109-110)

We assessed for differences in utilization in the preconception, prenatal and postpartum periods using analysis of covariance (ANCOVA) allowing us to adjust for maternal age, parity, rurality, history of cesarean delivery, hypertensive disease of pregnancy and diabetes during pregnancy.

1. Daw JR, Hatfield LA, Swartz K, Sommers BD. Women In The United States Experience High Rates Of Coverage ‘Churn’ In Months Before And After Childbirth. Health Affairs. 2017 Apr 1;36(4):598–606. 

2. Medicaid and CHIP Payment and Access Commission. Medicaid’s Role in Financing Maternity Care [Internet]. MACPAC; 2020 [cited 2022 May 23]. Available from: https://www.macpac.gov/wp-content/uploads/2020/01/Medicaid%E2%80%99s-Role-in-Financing-Maternity-Care.pdf

---

## [Decision Letter · Decision Letter 1]

21 Nov 2023

PONE-D-23-04247R1Not all Medicaid for pregnancy care is delivered equallyPLOS ONE

Dear Dr. Swartz,

Thank you for submitting your manuscript to PLOS ONE. After careful consideration, we feel that it has merit but does not fully meet PLOS ONE’s publication criteria as it currently stands. Therefore, we invite you to submit a revised version of the manuscript that addresses the points raised during the review process.

**ACADEMIC EDITOR: **There are remaining comments that reviewers feel not fulled addressed. ==============================

We look forward to receiving your revised manuscript.

Kind regards,

Kevin Lu, PhD

Academic Editor

PLOS ONE

Journal Requirements:

Reviewers' comments:

Reviewer's Responses to Questions

**Comments to the Author**

1. If the authors have adequately addressed your comments raised in a previous round of review and you feel that this manuscript is now acceptable for publication, you may indicate that here to bypass the “Comments to the Author” section, enter your conflict of interest statement in the “Confidential to Editor” section, and submit your "Accept" recommendation.

Reviewer #1: (No Response)

2. Is the manuscript technically sound, and do the data support the conclusions?

Reviewer #1: Partly

3. Has the statistical analysis been performed appropriately and rigorously? 

Reviewer #1: I Don't Know

4. Have the authors made all data underlying the findings in their manuscript fully available?

Reviewer #1: No

5. Is the manuscript presented in an intelligible fashion and written in standard English?

Reviewer #1: Yes

6. Review Comments to the Author

Reviewer #1: I thank the reviewers for revisions. I think some more could be done to strengthen the relevance and impact of the study conclusions:

(1) It is unclear to me why the authors are not using the prenatal utilization data from the birth record since they could observe non-Medicaid financed prenatal care. It can be challenging to measure prenatal care in claims due to bundled maternity billing so the birth record seems like an important and overlooked advantage. Instead of adding a limitation, they should address this using their data.

(2) I think the results would be more interpretable if they also reported the percent of the cohort with a preconception visit, adequate prenatal care, and postpartum visit since these will be less affected by outliers. The number of visits themselves seem less relevant since in the preconception and postpartum periods clinical recommendations are for one visit for most people.

(3) It says emergency Medicaid recipients were excluded but they are in Table 1. Also why exclude them when the goal is to compare across eligibility categories? They are an important and understudied group, so I don’t see the rationale in a descriptive study.

(4) Since all the outcomes are dependent on enrollment and the authors have access to enrollment data, I think it is critical that they report on mean months of enrollment during each of the time periods to contextualize their findings. If they wanted to go further, they could estimate models where they control for enrollment length to see if that is the primary explanatory variable of the differences in outcomes across eligibility groups.

(5) I think that the main policy takeaway of this study is missing -- it needs to be made much more clear in the discussion and abstract that these enrollment patterns are driven by eligibility income limits for pregnancy vs other groups which create discontinuities in insurance before versus during pregnancy. I think more discussion of how these limits vary across states and potential policy recommendations to smooth coverage in light of the findings would greatly strengthen the paper.

(6) I also think it is worth noting the fact that despite the fact that people are pregnant and in Medicaid, they are not enrolled in the pregnancy pathway when in theory, they should be. What are the possible implications of this for policy and for researchers who may rely on these categories to define pregnant enrollees?

(7) To align with typical language used by Medicaid programs, I would refer to it as “eligibility category” not enrollment category.

7. PLOS authors have the option to publish the peer review history of their article (what does this mean?). If published, this will include your full peer review and any attached files.

Reviewer #1: No

---

## [Author Response · Author response to Decision Letter 1]

8 Dec 2023

Response to reviewers – 12.8.23

Reviewer #1: I thank the reviewers for revisions. I think some more could be done to strengthen the relevance and impact of the study conclusions:

(1) It is unclear to me why the authors are not using the prenatal utilization data from the birth record since they could observe non-Medicaid financed prenatal care. It can be challenging to measure prenatal care in claims due to bundled maternity billing so the birth record seems like an important and overlooked advantage. Instead of adding a limitation, they should address this using their data.

Thank you for your thoughtful review. While we agree that there are limitations to claims data for identification of prenatal visits, the advantage is that we were able to report the timing of visits during a pregnancy. In contrast, birth certificate data would only report the total number of visits. So, while this might improve capture of prenatal visits that were not identified because of bundled billing, we would be unable to determine when visits occurred. Additionally, use of claims allows us to use consistent methodology in identification of preconception and postpartum visits which would not be included in birth certificate data.

We added an explanation to the methods (Line 100-102)

We used Medicaid claims rather than birth certificate data to identify prenatal visits because birth certificates report only the total number of visits, not the timing of visits during the pregnancy.

And to the limitations Line 203-204

Additionally, because maternity care is often bundled in global billing for delivery, we may have under-identified prenatal visits.

(2) I think the results would be more interpretable if they also reported the percent of the cohort with a preconception visit, adequate prenatal care, and postpartum visit since these will be less affected by outliers. The number of visits themselves seem less relevant since in the preconception and postpartum periods clinical recommendations are for one visit for most people.

We agree with the reviewer that examining these variables in a binary fashion to indicate whether care did or did not meet recommendations adds to the strength of our results. We conducted this analysis as a sensitivity analysis included in a supplemental table and discuss it in the results section. (Line 135-140) 

Results were similar in a sensitivity analysis conducted with dichotomized indicators of preconception, prenatal and postpartum care demonstrating a higher percentage of patients with traditional Medicaid had at least one-preconception visit than those with pregnancy Medicaid in both states. (Supplemental Table 1) A higher proportion of traditional Medicaid recipients also had at least seven prenatal visits than those with pregnancy Medicaid in both states. 

(3) It says emergency Medicaid recipients were excluded but they are in Table 1. Also why exclude them when the goal is to compare across eligibility categories? They are an important and understudied group, so I don’t see the rationale in a descriptive study.

We agree that this is an important and understudied population. We include them in Table 1 as an important component of the birthing population. We decided not to include them in other portions of the analysis as it was harder to make apples-to-apples comparisons with such a different population. Access to care for this population is largely determined by a different policy mechanism, state adoption of the CHIP Unborn Child provision allowing coverage of prenatal care. This has been studied extensively, including in these two states (Oregon provides prenatal care coverage, South Carolina does not), with greater focus on key metrics of access to care and health outcomes. Additionally, Emergency Medicaid recipients often rely on non-Medicaid-funded care for prenatal visits, for example, which would not have been captured in this dataset. 

(4) Since all the outcomes are dependent on enrollment and the authors have access to enrollment data, I think it is critical that they report on mean months of enrollment during each of the time periods to contextualize their findings. If they wanted to go further, they could estimate models where they control for enrollment length to see if that is the primary explanatory variable of the differences in outcomes across eligibility groups.

We agree with the reviewer that enrollment and utilization are correlated. We report the mean number of months of enrollment in the preconception, prenatal and postpartum periods in Supplemental Table 1. We did test our models controlling for enrollment and when we did so, our models produced errors due to collinearity between enrollment and utilization. When considering the relative importance of enrollment versus utilization, we opt to emphasize utilization as enrollment seems like an intermediate outcome compared to measured utilization of care. In Methods lines 104-105.

We also measured the number of months of enrollment at each perinatal time point.

In results lines 154-158

In Oregon, pregnancy and traditional Medicaid recipients had similar mean months of enrollment in the preconception (pregnancy 1.2 months vs traditional 1.5 months), prenatal (pregnancy 8.9 months vs traditional 9.1 months) and 61 day to 365 day postpartum periods (pregnancy 7.4 months vs traditional 8.0 months). In South Carolina, pregnancy Medicaid recipients had fewer mean months of enrollment in the preconception (pregnancy 0.5 months vs 1.5 months), prenatal (pregnancy 7.3 months vs traditional 10.4 months) and 61 day to 365 day postpartum periods (pregnancy 5.2 months vs traditional 8.0 months). (Supplemental Table 1)

(5) I think that the main policy takeaway of this study is missing -- it needs to be made much more clear in the discussion and abstract that these enrollment patterns are driven by eligibility income limits for pregnancy vs other groups which create discontinuities in insurance before versus during pregnancy. I think more discussion of how these limits vary across states and potential policy recommendations to smooth coverage in light of the findings would greatly strengthen the paper.

We expanded the discussion and conclusion including the points suggested by the reviewer, lines 205-228. We hope this brings this work better into the context of what is known on the topic and highlights how changes, especially those in the American Recovery Plan, may affect care.

(6) I also think it is worth noting the fact that despite the fact that people are pregnant and in Medicaid, they are not enrolled in the pregnancy pathway when in theory, they should be. What are the possible implications of this for policy and for researchers who may rely on these categories to define pregnant enrollees?

We have added an explanation for why it may be beneficial for Medicaid beneficiaries to be enrolled in traditional as opposed to pregnancy Medicaid. Regarding identification of pregnancies, the authors are not aware of research using only the pregnancy Medicaid category for identification of pregnancies. Rather, most research we are aware of uses an algorithm similar to that utilized in this paper starting with diagnosis, procedure and DRG codes. Lines 212-215.

Traditional Medicaid may be a favorable program for pregnant Medicaid beneficiaries because they are not at risk of losing eligibility after pregnancy. Additionally, some states limit the scope of covered services under pregnancy Medicaid as compared to traditional Medcaid plans.(11)

(7) To align with typical language used by Medicaid programs, I would refer to it as “eligibility category” not enrollment category.

We have updated throughout when discussing eligibility categories.

We have updated throughout when discussing eligibility categories.

---

## [Decision Letter · Decision Letter 2]

1 Feb 2024

PONE-D-23-04247R2Not all Medicaid for pregnancy care is delivered equallyPLOS ONE

Dear Dr. Swartz,

Thank you for submitting your manuscript to PLOS ONE. After careful consideration, we feel that it has merit but does not fully meet PLOS ONE’s publication criteria as it currently stands. Therefore, we invite you to submit a revised version of the manuscript that addresses the points raised during the review process. Please submit your revised manuscript by Mar 17 2024 11:59PM. If you will need more time than this to complete your revisions, please reply to this message or contact the journal office at plosone@plos.org. Please include the following items when submitting your revised manuscript:A rebuttal letter that responds to each point raised by the academic editor and reviewer(s). You should upload this letter as a separate file labeled 'Response to Reviewers'.A marked-up copy of your manuscript that highlights changes made to the original version. You should upload this as a separate file labeled 'Revised Manuscript with Track Changes'.An unmarked version of your revised paper without tracked changes. You should upload this as a separate file labeled 'Manuscript'.If applicable, we recommend that you deposit your laboratory protocols in protocols.io to enhance the reproducibility of your results. Protocols.io assigns your protocol its own identifier (DOI) so that it can be cited independently in the future. For instructions see: https://journals.plos.org/plosone/s/submission-guidelines#loc-laboratory-protocols. Additionally, PLOS ONE offers an option for publishing peer-reviewed Lab Protocol articles, which describe protocols hosted on protocols.io. Read more information on sharing protocols at https://plos.org/protocols?utm_medium=editorial-email&utm_source=authorletters&utm_campaign=protocols.

We look forward to receiving your revised manuscript.

Kind regards,

Bettye A. Apenteng

Academic Editor

PLOS ONE

Journal Requirements:

**Additional Editor Comments:**

Thank you for addressing most of the reviewers' comments. However, Reviewer#3 notes some important areas to address that will significantly improve the paper. I anticipate that these will require minor revisions to your current draft. 

Reviewers' comments:

Reviewer's Responses to Questions

**Comments to the Author**

1. If the authors have adequately addressed your comments raised in a previous round of review and you feel that this manuscript is now acceptable for publication, you may indicate that here to bypass the “Comments to the Author” section, enter your conflict of interest statement in the “Confidential to Editor” section, and submit your "Accept" recommendation.

Reviewer #1: All comments have been addressed

Reviewer #3: (No Response)

2. Is the manuscript technically sound, and do the data support the conclusions?

Reviewer #1: Yes

Reviewer #3: Partly

3. Has the statistical analysis been performed appropriately and rigorously? 

Reviewer #1: Yes

Reviewer #3: No

4. Have the authors made all data underlying the findings in their manuscript fully available?

Reviewer #1: No

Reviewer #3: No

5. Is the manuscript presented in an intelligible fashion and written in standard English?

Reviewer #1: Yes

Reviewer #3: Yes

6. Review Comments to the Author

Reviewer #1: This is making me write 100 characters in order to be able to move past this screen so I am writing in this box

Reviewer #3: 1. Abstract, reporting as 4.5 fold decrease implies a 22% relative risk of preconception visits for Pregnancy Medicaid vs traditional Medicaid. I think this is a more standard way of presenting the findings and it makes it easier for me to understand the magnitudes. Line 135-137 reports a 4.5:1 ratio of visits for TM beneficiaries --- this is easier to interpret as well.

2. Traditional Medicaid in South Carolina is a very different category than traditional Medicaid in Oregon, assuming that traditional Medicaid includes ACA Medicaid expansion to 138% FPL. Please be clear here.

3. I am having a hard time conceptualizing the rate as individuals with Pregnancy Medicaid only during this time period are only eligible for 60 days post-partum --- is there a differential rate in visits during this time period by Medicaid eligibility type or is most of the difference in visit rate after Day 60. Anderson et al PMID 35508289 has a good discussion on postpartum eligibility churn in North Carolina. How often are members switching categories of eligibility during the study period?

4. What is the average week of initial pre-natal visit by category? I am curious if the results are being driven by both right and left hand censoring of the Pregnancy Medicaid population?

Minor comment:

a) Lines 75-77 “majority of states” is that accurate, or is it better to say “all states…” as looking at the Kaiser Family Foundation data, all states have higher Pregnancy income thresholds than other categories of eligibility thresholds.

b) Lines 163-165 --- is it better expressed to say “traditional Medicaid beneficiaries had an average of 29% more visits than pregnancy Medicaid beneficiaries” as the current sentence seems to imply a 29% increase in the probability of at least one visit by TM but the supporting data indicates a count difference. Same Lines 165-167 for South Carolina

c) Figure 1 is hard to interpret as it seems that each panel resets to zero multiple times for each bar and the color distinctions are not clear and obvious. Perhaps a stacked bar chart might work better for each state?

7. PLOS authors have the option to publish the peer review history of their article (what does this mean?). If published, this will include your full peer review and any attached files.

Reviewer #1: No

Reviewer #3: No

---

## [Author Response · Author response to Decision Letter 2]

7 Feb 2024

Dear editors of PLOS One,

Thank you for the opportunity to revise “Not all Medicaid for pregnancy care is delivered equally” for a third time. We have responded below to the reviewer’s feedback below. 

Sincerely

Jonas Swartz, MD, MPH

Reviewer #1: This is making me write 100 characters in order to be able to move past this screen so I am writing in this box

Reviewer #3: 1. Abstract, reporting as 4.5 fold decrease implies a 22% relative risk of preconception visits for Pregnancy Medicaid vs traditional Medicaid. I think this is a more standard way of presenting the findings and it makes it easier for me to understand the magnitudes. Line 135-137 reports a 4.5:1 ratio of visits for TM beneficiaries --- this is easier to interpret as well.

We appreciate the reviewers feedback that our reporting was unclear. As this is a continuous measure, we opted to use what the reviewer suggested is easier to interpret language from the body of the paper (modified per reviewer recommendation in later comments. 

“Discussion: In South Carolina, those using traditional Medicaid had 450% more preconception visits and 70% more postpartum visits compared with pregnancy Medicaid. In Oregon, those using traditional Medicaid had 200% more preconception visits and 29% more postpartum visits than individuals using pregnancy Medicaid.”(Line 59-63)

2. Traditional Medicaid in South Carolina is a very different category than traditional Medicaid in Oregon, assuming that traditional Medicaid includes ACA Medicaid expansion to 138% FPL. Please be clear here.

We stipulate specific eligibility thresholds in the Methods, line 102-104. Further discussed on lines 209-216.

In Oregon, adults earning up to 138% of the Federal Poverty Level (FPL) may enroll in traditional Medicaid, while in South Carolina, they may earn up to 67% of FPL.

3. I am having a hard time conceptualizing the rate as individuals with Pregnancy Medicaid only during this time period are only eligible for 60 days post-partum --- is there a differential rate in visits during this time period by Medicaid eligibility type or is most of the difference in visit rate after Day 60. Anderson et al PMID 35508289 has a good discussion on postpartum eligibility churn in North Carolina. How often are members switching categories of eligibility during the study period?

We acknowledge that one limitation of this analysis is that care may occur with payors other than Medicaid and not be captured in claims data. It is possible that those with pregnancy Medicaid who lose coverage after 60 days subsequently seek care elsewhere. This analysis does stratify by timing of postpartum care visits and we report, in Table 2, care up to 60 days and care from 61-365 days. We hope this addresses the reviewer’s first concern. Regarding churn, thank you for this excellent citation. We added this concept and reference to the discussion. (Line 232-233) The reviewer’s final question about members switching categories of eligibility during the study period is good question that is out of the scope of this analysis but would be an interesting question for a future study. 

“Data from North Carolina indicate a larger proportion of those with pregnancy Medicaid were at risk of losing coverage by four months postpartum.(11)”

4. What is the average week of initial pre-natal visit by category? I am curious if the results are being driven by both right and left hand censoring of the Pregnancy Medicaid population?

The week of initiation of prenatal care by category would be an interesting question for future study. The reviewer is correct that the Pregnancy Medicaid population is less likely to be enrolled and thus less likely to access care before enrollment and after 60 days postpartum. We hoped with this manuscript to illustrate what that looked like from a utilization standpoint (number of visits). (Lines 212-214)

“We also did not analyze timing of initiation of prenatal care, though that would be an opportunity for future research.”

Minor comment:

a) Lines 75-77 “majority of states” is that accurate, or is it better to say “all states…” as looking at the Kaiser Family Foundation data, all states have higher Pregnancy income thresholds than other categories of eligibility thresholds.

South Dakota has the same threshold for traditional Medicaid and pregnancy. We have made the language more specific as requested. (Lines 84-86)

“States have multiple categories of Medicaid programs enrolling pregnant people. Recognizing the importance of obstetrical care, all states but South Dakota cover pregnant people through pregnancy Medicaid at higher income thresholds than non-pregnant adults.”

b) Lines 163-165 --- is it better expressed to say “traditional Medicaid beneficiaries had an average of 29% more visits than pregnancy Medicaid beneficiaries” as the current sentence seems to imply a 29% increase in the probability of at least one visit by TM but the supporting data indicates a count difference. Same Lines 165-167 for South Carolina

We adjusted the language in the discussion and abstract as suggested by the reviewer. (line 180-185)

Care access postpartum was also different depending on Medicaid eligibility, especially from 61-365 days postpartum. In Oregon those with traditional Medicaid had 29% more 61-365 days postpartum as those with pregnancy Medicaid (mean 8.3 visits (SE 0.09) vs 6.3 visits (SE 0.08)). In South Carolina, those with traditional Medicaid had 70% more visits 61-365 days postpartum as those with pregnancy Medicaid (mean 4.8 visits (SE 0.04) vs mean 2.3 visits (SE 0.03)).

c) Figure 1 is hard to interpret as it seems that each panel resets to zero multiple times for each bar and the color distinctions are not clear and obvious. Perhaps a stacked bar chart might work better for each state?

We appreciate this feedback about the interpretability of this data visualization. We prefer to keep this heatmap format. We have updated the figure legend to provide a more detailed explanation.

“Legend: Outpatient visits among two Medicaid enrollment categories in Oregon (panel a) and South Carolina (panel b). Dashed lines demarcate the preconception period, delivery and 60 days after delivery as, during the study period, pregnancy Medicaid coverage ended at this time postpartum. Each vertical rectangle in the panel represents average number of outpatient visits per person per month. Darker color indicates a higher number of outpatient visits during that month.”

---

## [Decision Letter · Decision Letter 3]

16 Feb 2024

Not all Medicaid for pregnancy care is delivered equally

PONE-D-23-04247R3

Dear Dr. Swartz,

We’re pleased to inform you that your manuscript has been judged scientifically suitable for publication and will be formally accepted for publication once it meets all outstanding technical requirements.

Kind regards,

Bettye A. Apenteng

Academic Editor

PLOS ONE

Additional Editor Comments (optional):

Reviewers' comments:

Reviewer's Responses to Questions

**Comments to the Author**

1. If the authors have adequately addressed your comments raised in a previous round of review and you feel that this manuscript is now acceptable for publication, you may indicate that here to bypass the “Comments to the Author” section, enter your conflict of interest statement in the “Confidential to Editor” section, and submit your "Accept" recommendation.

Reviewer #3: All comments have been addressed

2. Is the manuscript technically sound, and do the data support the conclusions?

Reviewer #3: Yes

3. Has the statistical analysis been performed appropriately and rigorously? 

Reviewer #3: Yes

4. Have the authors made all data underlying the findings in their manuscript fully available?

Reviewer #3: (No Response)

5. Is the manuscript presented in an intelligible fashion and written in standard English?

Reviewer #3: Yes

6. Review Comments to the Author

Reviewer #3: I am still not a huge fan of the data presentation in Figure 1 but the writing is clear and the primary points of contention have been addressed.

Nice work!

7. PLOS authors have the option to publish the peer review history of their article (what does this mean?). If published, this will include your full peer review and any attached files.

Reviewer #3: No

---

## [Editor Report · Acceptance letter]

25 Mar 2024

PONE-D-23-04247R3 

PLOS ONE

Dear Dr. Swartz, 

I'm pleased to inform you that your manuscript has been deemed suitable for publication in PLOS ONE. Congratulations! Your manuscript is now being handed over to our production team.

Kind regards, 

on behalf of

Dr. Bettye A. Apenteng 

Academic Editor

PLOS ONE